# In Vitro Comparative Study of Platelets Treated with Two Pathogen-Inactivation Methods to Extend Shelf Life to 7 Days

**DOI:** 10.3390/pathogens11030343

**Published:** 2022-03-11

**Authors:** Nicolas Malvaux, Fanette Defraigne, Styliani Bartziali, Camille Bellora, Kathleen Mommaerts, Fay Betsou, Anne Schuhmacher

**Affiliations:** 1Red Cross of Luxemburg, Boulevard Joseph II, 40, L-1840 Luxembourg, Luxembourg; fanette.defraigne@croix-rouge.lu (F.D.); styliani.bartziali@croix-rouge.lu (S.B.); anne.schuhmacher@croix-rouge.lu (A.S.); 2Integrated Biobank of Luxembourg, 1 rue Louis Rech, L-3555 Dudelange, Luxembourg; camille.bellora@ibbl.lu (C.B.); kathleen.mommaerts@ibbl.lu (K.M.); fay.betsou@lns.etat.lu (F.B.); 3Luxembourg Center for Systems Biomedicine, 6 Av. du Swing, L-4367 Esch-sur-Alzette, Luxembourg; 4Laboratoire National de Sante, 1 rue Louis Rech, L-3555 Dudelange, Luxembourg

**Keywords:** 7-day platelet storage, pathogen inactivation, platelet quality

## Abstract

Background and Objectives: Since 2015, platelet products have been pathogen-inactivated (PI) at the Luxemburgish Red Cross (LRC) using Riboflavin and UV light (RF-PI). As the LRC should respond to hospital needs at any time, platelet production exceeds the demand, generating a discard rate of 18%. To reduce this, we consider the extension of storage time from 5 to 7 days. This study’s objective was to evaluate the in vitro 7-day platelet-storage quality, comparing two PI technologies, RF-PI and amotosalen/UVA light (AM-PI), for platelet pools from whole-blood donations (PPCs) and apheresis platelets collected from single apheresis donation (APCs). Materials and Methods: For each product type, 6 double-platelet concentrates were prepared and divided into 2 units; one was treated with RF-PI and the other by AM-PI. In vitro platelet-quality parameters were tested pre- and post-PI, at days 5 and 7. Results: Treatment and storage lesions were observed in PPCs and APCs with both PI methods. We found a higher rate of lactate increase and glucose depletion, suggesting a stronger stimulation of the glycolytic pathway, a higher Annexin V binding, and a loss of swirling in the RF-PI-treated units from day 5. The platelet loss was significantly higher in the AM-PI compared with the RF-PI units. Conclusions: Results suggest that RF-PI treatment has a higher deleterious impact on in vitro platelet quality compared to AM-PI, but we observed higher loss of platelets with AM-PI due to the post-illumination amotosalen adsorption step. If 7-day storage is needed, it can only be achieved with AM-PI, based on our quality criteria.

## 1. Introduction

Pathogen inactivation (PI) of platelet (PLT) concentrates (PCs) increases the safety of blood products by reducing the risk of transfusion-transmitted infections [1] and by inactivating leukocytes for the prevention of transfusion-associated graft-versus-host disease (TA-GVHD) [2,3]. Two inactivation methods are commercially available. One of them uses a photodynamic process with Riboflavin and UV A/B, damaging nucleic acids using photosensitized riboflavin and by the release of oxygen radicals (MIRASOL System, Terumo BCT, RF-PI). The second technology uses a photochemical reaction with amotosalen, irreversibly crosslinking nucleic acids in the presence of UVA light and preventing replication and transcription (INTERCEPT Blood System, Cerus, AM-PI). These treatments may also contribute to platelet-storage lesions (PSL) [4,5], including PLT activation, degranulation, protein modifications and release, and extracellular milieu composition changes, pro-apoptotic signaling, and morphological changes. These do not necessarily impact clot formation [6] or clinical outcome [7].

Since 2015, the Red Cross of Luxemburg (LRC) has been using RF-PI, chosen primarily because of its ease of use and short delays in product availability.

As the LRC should ensure self-sufficiency and be able to respond to hospital needs at any time, platelet production systematically exceeds average demand, generating a discard rate of outdated PCs as high as 24% of pooled platelet and 5% of apheresis platelet products in 2020. To reduce this, we set out to validate the extension of storage time from 5 to 7 days, comparing RF-PI to AM-PI. The inactivation methods have already been characterized for their pathogen-inactivation profile, induced PSL, transfusion safety, and efficacy [8]. 

A recent study performed by McDonald and colleagues [9] showed that bacterial inactivation capability of AM-PI was significantly greater than that of RF-PI at the end of a 7-day shelf life. 

The hemostatic efficacy of pathogen-inactivated vs untreated platelets was assessed in several clinical trials for AM-PI with up to 5-day [10,11] or 7-day storage [12], and for RF-PI with 5-day shelf life [13,14,15]. A single-arm study investigated the clinical performance of RF-PI with CCI as a primary outcome with up to 7-day storage [16]. Recent studies investigated platelet use in real-life conditions with up to 7-day storage for AM-PI [17,18], and did not show a significantly increased consumption. A routine evaluation study was also conducted with RF-PI with 7-day maximum shelf life, showing an increased use balanced by a reduction in discards [19]. 

A few comparative studies [20,21,22] have been conducted to evaluate the PSL induced by AM-PI and RF-PI upon storage for at least 7 days. In the present study, critical in vitro properties of 7-day stored platelets, previously treated with AM-PI or RF-PI, were examined. Apheresis concentrates (APCs) and platelet pools (PPCs) derived from whole blood and prepared with a Reveos automated blood-processing system (Terumo BCT) were considered. 

We adopted an experimental design permitting paired comparisons from the same pre-treatment products to limit inter-individual and pre-analytical variability. 

## 2. Results

### 2.1. PLT Product Quality Attributes

All the platelet concentrates conformed to the specifications of the RF-PI and AM-PI treatments, in terms of volume, platelet concentration, and platelet dose. The platelet contents met the European Directorate for the Quality of Medicines (EDQM) [23] guidelines throughout the storage time in both types of PCs and with both treatments. The platelet concentration was consistently higher in AM-PI than RF-PI platelet products, but the platelet dose was lower (Table 1). For PPCs, the average platelet loss with processing was 1.5% for RF-PI and 5.3% for AM-PI, while for APCs, it was 5.8% and 12%, respectively. 

### 2.2. PLT Storage Study

In all platelet products, anaerobic glycolysis generated a decrease in the glucose concentration and corresponding lactate production, which was more pronounced in the RF-PI units. The difference in the AM-PI units was statistically significant from day 5; see Figure 1. 

In PPCs, the pO_2_ was quite stable until day 5 with both PI treatments. This was also found in the APCs, but the pO_2_ in the RF-PI-treated products was significantly lower than in the AM-PI-treated units. The pCO_2_ had a reductioFDn tendency between day 5 and day 7, and was significantly lower in RF-PI PCs. Although the pH was relatively stable in all AM-PI units, we found significant pH decrease in PPs and APCs subject to RF-PI treatment from day 5 onwards; see Figure 2.

PLT activation, measured by the soluble P-Selectin concentration in the supernatant, was not significantly different between RF-PI and AM-PI in APCs at day 7. On the contrary, the soluble p-Selectin in AM-PI PPs was significantly higher than in RF-PI PPCs at day 7. Regarding the phosphatidylserine expression, all RF-PI-treated PPs showed a higher Annexin V binding level than AM-PI from day 2 onwards. This was also found for APCs at day 7; see Figure 3.

The LDH release did not show significant differences throughout the storage time in either PPs or APCs. Nevertheless, a significant difference between RF-PI and AM-PI-treated units was found only at day 5 for APCs, with higher LDH release in AM-PI.

Although swirling was maintained in all AM-PI units, we found a drop of the swirling index at day 5 for the RF-PI units, with a complete disappearance in all the APC samples. RF-PI units had higher MPV values than the AM-PI units, for both types of PC treatments, from day 5 onwards; see Table 2.

## 3. Discussion

PI methods have been shown to reduce the risk of transfusion-transmitted infections by platelet products, and can effectively replace gamma irradiation for the prevention of transfusion-associated grafts versus host disease. The bacterial growth in platelets stored at room temperature limits their shelf life to 5 days. A study by Gorria et al. [24] showed that a 2-day extension to the shelf life of PCs results in reductions in outdates ranging from 88.4% to 100%. Our discard rate, using a 5-day expiry time, has been 18% for combined whole-blood-derived and apheresis PCs. We considered that extending the shelf life to 7 days could reduce the outdated product rate and avoid shortages. 

The deleterious effects of PI on platelet quality have been shown in several studies, including platelet activation [25,26,27], induction of pro-apoptotic signaling pathways [28,29], enhancement of platelet glycolysis, and irreversible oxidation of proteins [30]. 

However, very few direct comparisons between PI methods exist [20,21,22]. In addition, while APCs and buffy coat PCs have been largely characterized, PPCs obtained using Reveos and subsequently PI-treated would merit further investigation [31]. 

The main objective of our study was to evaluate the impact of RF-PI and AM-PI on the critical properties of stored platelets to validate a method permitting prolonged storage in routine operation.

The platelet concentration was higher for AM-PI than RF-PI due to the lower volume of amotosalen versus riboflavin solution added to the PCs. However, the platelet loss associated with processing was higher for AM-PI, due to the retention of platelets during the CAD step. 

Similar to other studies [22,27,32,33], our results showed that platelet metabolism was affected by both processing methods combined with storage. The decrease of the glucose concentration and the increase of the lactate production resulting from glycolysis [34] were more pronounced in the RF-PI-treated platelet products. The exhaustion of glucose and the excess of lactate are associated with a decrease in energy levels and an increase in the markers of cell death [35]. It is also well established that lactate accumulation in the storage solution causes metabolic acidosis and contributes to a decline of PLT quality overall [36,37].

PS expression, which has been demonstrated to be a relevant pro-apoptotic marker [38], showed an increase in all PCs processed by both methods, throughout the storage time, as previously reported by others [26,29,39]. For the AM-PI treatment, published data have been controversial [40]. Even if the increase in PS expression is a reproducible phenomenon occurring throughout storage [41], and despite the absence of the untreated control in the study, we suggest that there is a treatment effect on this parameter. Indeed, our results show higher PS expression in the RF-PI from day 5 in PPCs and at day 7 in APCs. The effect of the RF-PI treatment on the advent of apoptosis has also been demonstrated in other studies [28,39], while no significant AM-PI effect was found by others [42].

A decrease throughout the storage time of the pCO_2_ was observed more specifically in RF-PI PCs and reinforces the hypothesis of a decline in the oxidative phosphorylation pathway and a switch to glycolysis. 

These findings are in agreement with Abonnenc et al. [21], but in contradiction with Picker et al., in a paired comparison study, performed on apheresis-derived platelets, treated with either RF-PI or AM-PI, they concluded a better maintenance of the respiratory mitochondrial function in RF-PI [22,32]. Their conclusions were based on a lower oxygen consumption rate in AM-PI, associated with a lower ATP content and combined with less cell viability, as demonstrated by the decreased hypotonic shock response, increased transmembrane mitochondrial potential, and a more pronounced Annexin A5 release. As the parameters studied and the methods used were different, we cannot explain these divergent observations. In our hands, the LDH release, which is considered to be an indicator of platelet disruption, was systematically higher in the AM-PI units, throughout the storage time. This moderate increase was also found in previous studies [30] and can be partially explained by the mechanical stress during the CAD step [21]. 

The difference in soluble CD 62P was generally not significant between platelets treated with the two PI methods. Nevertheless, a trend toward higher values was observed from day 5 onwards in AM-PI products. This difference in platelet activation has also been highlighted in review papers [4,25]. Other studies did not show a significant platelet activation of AM-PI beyond the storage effect [43,44].

The preservation of the platelet discoid shape can be assessed by examining the product through a light source. This assessment is systematically performed before product release at the LRC. In its absence, PPCs are not distributed and discarded. Swirling intensity depends on platelet discoid morphology [45,46]. We found a decrease in the swirling index from day 5 in all types of RF-PI products. Swirling completely disappeared in the APCs from day 5, while it was maintained in all AM-PI products. Poor swirling is correlated with low pH values and loss of in vivo viability [45]. It is highly predictive of poor post-transfusion platelet count increments, and increases the risk of a transfusion reaction [47]. The disappearance of the platelet swirl was also reported by the Belgian Red Cross in Flanders [5] in more than 10% of their PPs, and was one of the reasons why RF-PI was not implemented. Regarding these results, and even if the assessment of the platelet swirl remains a subjective operator-dependent quality parameter [48], and considering that the LRC will maintain this release criterion, RF-PI could not be distributed after day 5, while this would still be possible for AM-PI up to day 7.

RF-PI units showed increased MPV values compared with AM-PI units for both types of PPs from day 5 onwards. Even if the MPV is not a perfect indicator of morphological change [45], it is consistent with the evolution of the platelet swirling scores.

In conclusion, this in vitro assessment of the storage of AM-PI and RF-PI PPs shows different metabolic and morphological impacts between the two methods. This is not uniformly dependent on the platelet preparation method for all quality attributes examined. A 7-day shelf-life extension appears not to be feasible with RF-PI, but could be achieved with AM-PI. The selection of a fit-for-purpose method is based on the different platelet-quality parameters, as well as operational aspects, particularly those related to incoming product specifications and treatment time constraints. Therefore, evaluation in routine conditions should also be performed following initial in vitro evaluation, reported here. 

## 4. Materials and Methods

### 4.1. Platelet Preparation

PPCs were obtained from 475 mL whole-blood collections in CPD anticoagulant, separated into components using the Reveos system (Terumo BCT, Lakewood, CO, USA). On day 1 and after a minimum of 4 h resting time, 5 interim platelet units (IPUs) were mixed and suspended in 250 mL of platelet additive solution (T-PAS+, Terumo BCT, Lakewood, CO, USA) to obtain a ratio of PAS/plasma volume of 62%.

Double-dose APCs were collected with a Trima Accel apheresis device (Terumo BCT, Lakewood, CO, USA) version 7 with automatic addition of approximatively 62% of T-PAS+. The target platelet content was 6.2–6.5 × 10^11^ platelets per bag. 

### 4.2. Study Design

For each type of platelet product (PPCs and APCs), two groups were considered—the PCs treated with RF-PI and the PCs treated with AM-PI.

For PPCs, immediately after the platelet preparation (day 1), two identical ABO group units were mixed and equally split. In total, 6 pairs of identical PPs were prepared, and for each pair, one product was connected (TSCD, Terumo BCT, Lakewood, CO, USA) to a RF-PI processing set (RF-PI, ref 10790, Terumo BCT, Lakewood, CO, USA) under sterile conditions, and the other one to an AM-PI processing set (AM-PI, ref Large Volume (LV) INT2204B, Cerus Europe BV, Amersfoort, The Netherlands), for the RF-PI or AM-PI treatment, respectively.

For APCs, after T-PAS+ addition and a minimum time of 1 h resting and 1 h agitation (day 0), double-dose platelet units were each divided into two identical products. One product was connected to a RF-PI processing set (RF-PI, ref 10790) under sterile conditions, the other one to an AM-PI processing set (AM-PI, ref Small Volume (SV INT2104), Cerus Europe BV, Amersfoort, The Netherlands) for the RF-PI or AM-PI treatment, respectively.

### 4.3. Pathogen-Inactivation Treatment

For AM-PI, each PC was treated with a nominal concentration of 150 μM amotosalen and 3.0 J/cm^2^ UVA light (320 to 400 nm) in an INTERCEPT illuminator (INT100, Cerus Europe BV, Amersfoort, Netherlands). Each product was then incubated for 16 h with a Compound Adsorption Device (CAD) to remove residual amotosalen and photoproducts. During the CAD step and storage, PLT units were stored under gentle agitation at a temperature of 22 ± 2 °C (Helmer, Noblesville, IN, USA).

For RF-PI, platelets were treated, after riboflavin addition (35 mL, 500 μM), in an ultraviolet A/B (280–400 nm) illuminator (Terumo BCT, Lakewood, CO, USA), with an exposure time depending on the PLT volume. After treatment, PCs were stored for 7 days under the same conditions. 

### 4.4. Sampling of Platelet Concentrates

For APCs, samples were taken on day 0 as baseline, then on day 1 after PI, and on day 5 and day 7 of storage. For PPCs, samples were taken on day 1 as baseline, on day 2 after PI, and on day 5 and day 7 of storage.

### 4.5. Analytical Methods

Mean Platelet Volume (MPV) and platelet concentration were analyzed automatically (DHX-900 Beckman–Coulter, Brea, CA, USA). Blood gases, pH, lactate, and glucose concentration were measured on platelet samples using an ISTAT analyzer (Abbott Diagnostics, Chicago, IL, USA).

Lactate Dehydrogenase (LDH) activity was evaluated on fresh platelet supernatants and lysates using the LDH Activity Assay Kit (Sigma Aldrich, Saint-Louis, MO, USA) and the Synergy MX spectrophotometer (BioTek, Winooski, VT, USA). 

Soluble P-selectin levels were measured in platelet supernatants using the Human P-Selectin/CD62P Quantikine ELISA Kit (R&D Systems, Minneapolis, MN, USA). 

Annexin V binding on the outer plasma membrane of the platelets was measured by fluorescence (FITC, Invitrogen Alexa, Eugene, OR, USA) using flow cytometry (Cytoflex, Beckman–Coulter, Brea, CA, USA). Swirling score was visually assessed using a three-level categorical scale: 2—good; 1—intermediate; 0—no swirl.

### 4.6. Statistics

Data were analyzed using descriptive statistical methods. Comparative analyses between the two treatments were performed using Minitab software version 20.

A variance equality analysis (Levene) was conducted. If this analysis did not show any significant difference between RF-PI and AM-PI, a paired *t*-test was performed at each time point.

## Figures and Tables

**Figure 1 pathogens-11-00343-f001:**
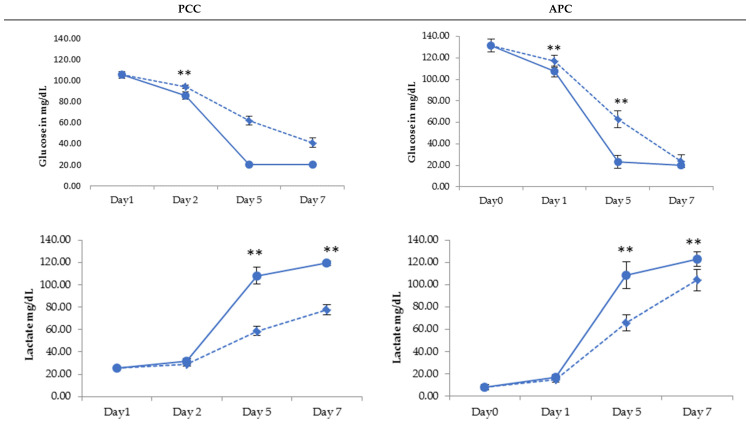
Evolution of glucose and lactate concentration evolution of platelets treated with INTERCEPT (AM-PI—dotted line) and MIRASOL (RF-PI—solid line) upon 7-day storage for PPCs (Platelet Pool Concentrates) and APCs (Apheresis Platelet Concentrates); * *p* < 0.05, ** *p* < 0.01.

**Figure 2 pathogens-11-00343-f002:**
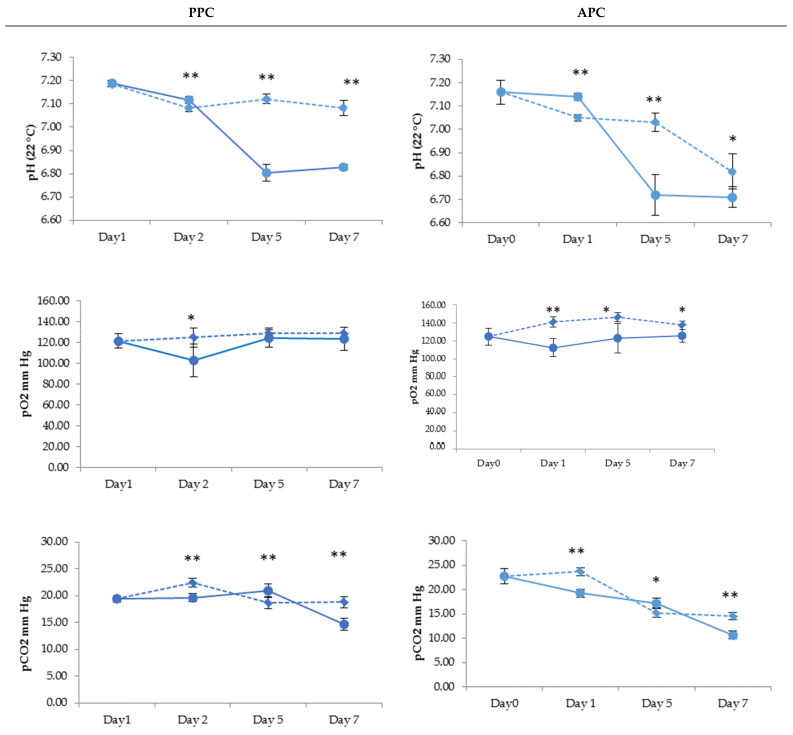
Evolution of pH, pCO2, and pO2 in platelets treated with INTERCEPT (AM-PI—dotted line) and MIRASOL (RF-PI—solid line) upon 7-day storage for PPCs (Platelet Pool Concentrates) and APCs (Apheresis Platelet Concentrates); * *p* < 0.05, ** *p* < 0.01.

**Figure 3 pathogens-11-00343-f003:**
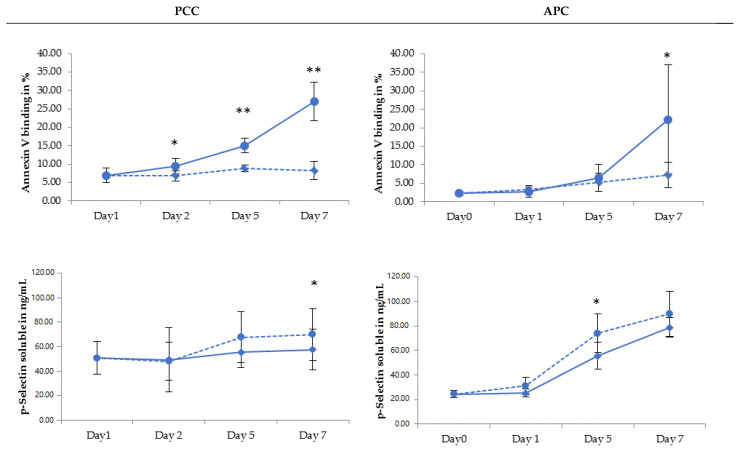
Evolution of apoptosis and platelet activation markers in platelets treated with INTERCEPT (AM-PI—dotted line) and MIRASOL (RF-PI—solid line) upon 7-day storage for PPCs (Platelet Pool Concentrates) and APCs (Apheresis Platelet Concentrates); * *p* < 0.05, ** *p* < 0.01.

**Table 1 pathogens-11-00343-t001:** Characteristics of PPCs (Platelet Pool Concentrates) and APCs (Apheresis Platelet Concentrates) prepared with INTERCEPT (AM-PI) and MIRASOL (RF-PI) and stored for up to 7 days. Mean and standard deviation values are shown.

	PPC				APC			
	Day 1 Before PI	Day 2	Day 5	Day 7	Day 0 Before PI	Day 1	Day 5	Day 7
Volume (mL)								
RF-PI	370 ± 3	397 ± 4	381 ± 5	369 ± 5	282 ± 5	305 ± 5	291 ± 5	282 ± 3
AM-PI	373 ± 6	360 ± 6	345 ± 6	335 ± 5	281 ± 5	260 ± 6	246 ± 7	237 ± 7
Platelet Concentration (10^9^/L)								
RF-PI	890 ± 54	818 ± 35	815 ± 48	771 ± 49	1107 ± 48	964 ± 33	988 ± 44	953 ± 62
AM-PI		874 ± 43	838 ± 35	814 ± 47		1052 ± 76	1016 ± 43	993 ± 54
*t*-Test		*p = 0.001*	ns	*p = 0.03*		*p = 0.012*	*ns*	*p = 0.013*
Platelet Dose (10^11^ per unit)								
RF-PI	3.30 ± 0.19	3.24 ± 0.12	3.10 ± 0.16	2.84 ± 0.17	3.13 ± 0.18	2.95 ± 0.13	2.87 ± 0.17	2.69 ± 0.20
AM-PI	3.32 ± 0.20	3.14 ± 0.19	2.89 ± 0.11	2.73 ± 0.15	3.11 ± 0.19	2.74 ± 0.25	2.50 ± 0.17	2.36 ± 0.17
*t*-Test		*p = 0.043*	*p = 0.03*	*ns*		*p = 0.017*	*p < 0.001*	*p < 0.001*

**Table 2 pathogens-11-00343-t002:** Platelet lysis and morphology indicator evolution in platelets treated with INTERCEPT (AM-PI—dotted line) and MIRASOL (RF-PI—solid line) upon 7-day storage for PPCs (Platelet Pool Concentrates) and APCs (Apheresis Platelet Concentrates).

	PPC				APC			
	Day 1 Before PI	Day 2	Day 5	Day 7	Day 0 Before PI	Day 1	Day 5	Day 7
LDH Release (%)								
RF-PI	4.74 ± 1.37	3.84 ± 0.38	3.96 ± 0.29	4.29 ± 0.43	2.34 ± 0.92	2.90 ± 1.15	2.84 ± 0.69	3.13 ± 0.43
AM-PI		3.80 ± 0.21	5.29 ± 1.21	5.43 ± 1.56		3.01 ± 0.61	3.67 ± 0.76	3.67 ± 0.44
*t*-Test		ns	ns	ns		ns	*p* = 0.002	ns
MPV								
RF-PI	8.06 ± 0.37	8.00 ± 0.33	8.92 ± 0.42	9.84 ± 0.49	7.15 ± 0.55	6.84 ± 0.56	8.11 ± 0.95	8.85 ± 1.07
AM-PI		8.11 ± 0.30	8.15 ± 0.28	8.23 ± 0.36		7.10 ± 0.53	7.53 ± 0.68	7.76 ± 0.59
*t*-Test		*p* = 0.004	*p* < 0.001	*p* < 0.001		*p* = 0.004	*p* = 0.011	*p* = 0.003
Swirl								
RF-PI	2.00	1.92	1.17	0.17	2.00	1.83	1.67	0.00
AM-PI		2.00	2.00	2.00		2.00	2.00	2.00

## Data Availability

Additional supporting information or data can be asked to the corresponding author.

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
