# Peer review of "In Vitro Comparative Study of Platelets Treated with Two Pathogen-Inactivation Methods to Extend Shelf Life to 7 Days"

_pathogens, 2022, doi:10.3390/pathogens11030343_

Round 1

Reviewer 1 Report

  1. The study compares apheresis platelets versus pooled platelets prepared with the Reveos system and stored for 7 days after treatment with two pathogen reduction technologies. The study found differences and similarities with the outcomes of other studies. 
  2. Manuscript style. Please use consistently PPC as defined in line 69 (not PPs, as in lines 106, 113, 114, 121, 199). I believe PPs in lines 205 and 230  should be PCs: please check.
  3. Manuscript style. I suggest to use the period (and not comma)  punctuation for numerical values in Tables 1 and 2 (English style). Please translate 'par poche' in Platelet Dose (Table 1).
  4. Manuscript style, Figure 1. Edit caption: 'Evolution of glucose and lactate evolution ...'. Please edit ordinate units of measurement: e.g. Glucose (mg/dl), Lactate (mg/dl), same for Fig 2 and 3. Use consistently Baseline Day 1 or Day 0 (not (Day 1) or (Day 0) in parentheses in Figures 1-3. In Figure 2 '(22°)' should be '(22 °C)'.
  5. Manuscript style, Figure 3. pSelectin should be p-Selectin.
  6. Lines 111-113. Authors report no significantly differences in p-Selectin between RF-PI and AM-PI in APC at day 7. Figure 3 bottom right panel confirms this statement. Please clarify the sentence: 'On the contrary, the soluble p-Selectin in AM-PI PPs (PPCs) was significantly higher than in RF-PI PPCs'. The visual appreciation from the bottom left panel of Figure 3 data on day 7 from AM and RF platelets looks quite similar to the right panel, with a large superimposition of the two data distributions. I would expect no significant difference also for PPCs.
  7. Manuscript style. Line 151. Spell out CAD which appears here for the first time. Do not spell it out again in lines 242-243. Lines 178-179. I believe 'increased' should be 'increase'. Lines 160, 163, 166. Please use consistently p-Selectin and not PS. Lines 161 and 171: PC's should be PCs.
  8. Materials and Methods. Please add City and Country for each manufacturer.
  9. Please add the Author contribution and Data availability statements.
  10. Reference 1 is quite old and does not include the most recent clinical investigations. I suggest to replace it with Escolar G, Diaz-Ricart M, McCullough J. Impact of different pathogen reduction technologies on the biochemistry, function, and clinical effectiveness of platelet concentrates: An updated view during a pandemic. Transfusion. 2022
    Jan;62(1):227-246. doi: 10.1111/trf.16747.

Author Response

Dear Reviewer, thank you for your comments, all these were taken in account and the manuscript has been modified in this way:

2) Modifications were done

3) Per poche was translated in "per unit"

4) Style was modified and adapted

5) p-Selectin was corrected

6) The error bars represent standard deviation only. T test result was p=0,049 which is very closed to the significance level

7) CAD was explained and spelled out at line 255

8) Done

9) Done

10) Done

Reviewer 2 Report

Authors considered extension of the storage time from 5 to 7 days. They evaluated the in vitro 7-day platelet storage quality, comparing two PI technologies, RF-PI and amotosalen/ UVA light (AM-PI) for platelet pools (PPCs) and apheresis platelets (APCs).

Although this manuscript is interesting and important in clinical practice, several issues arise.

Abstract

What was the difference between PPC and APC?

Did authors recommend the AM-PI technique?

Line 25. The platelet loss was significantly higher in the AM-PI compared with the RF-PI units. Please show raw data.

Table 1

Abbreviations should be explained in legends.

Figures 1-3

Statistical analyses may be useful.

Table 2

There were no dotted or solid lines.

What is “Swirl”? Please pxplain.

Abbreviations should be explained in legends.

Minor points

Abbreviations such as PPC, APC, RF-PI and AM-PI should be explained in legends of Tables.

Author Response

Dear Reviewer, thank you for your comments, all these were taken in account and the manuscript has been modified in this way:

> More explanations were added in the abstract part

> Last sentence was adapted

> Please find raw data and results of the T TEST in attachment (Comparative test for PLT Loss)

> Table 1 : abbrevations were added

> Statistics were added on the graphs with comment in legend : p< 0,05 * and p < 0,01 **

> Table 2 lines were modified

> swirl was explained in line 200

> Abbrevations were explained in legends
